# ABSTRACTORS AND RELATIONAL CROSS-ATTENTION: AN INDUCTIVE BIAS FOR EXPLICIT RELATIONAL REASONING IN TRANSFORMERS

**Awni Altabaa[1], Taylor Webb[2], Jonathan D. Cohen[3], John Lafferty[1]**
[1]Yale University, [2]UCLA, [3]Princeton University

## ABSTRACT

An extension of Transformers is proposed that enables explicit relational reasoning through a novel module called the *Abstractor*. At the core of the Abstractor is a variant of attention called *relational cross-attention*. The approach is motivated by an architectural inductive bias for relational learning that disentangles relational information from object-level features. This enables explicit relational reasoning, supporting abstraction and generalization from limited data. The Abstractor is first evaluated on simple discriminative relational tasks and compared to existing relational architectures. Next, the Abstractor is evaluated on purely relational sequence-to-sequence tasks, where dramatic improvements are seen in sample efficiency compared to standard Transformers. Finally, Abstractors are evaluated on a collection of tasks based on mathematical problem solving, where consistent improvements in performance and sample efficiency are observed.

## 1 INTRODUCTION

The ability to infer and process relations and reason in terms of analogies lies at the heart of human abilities for abstraction and creative thinking (Snow et al., 1984; Holyoak, 2012). This capability is largely separate from our ability to acquire semantic and procedural knowledge through sensory tasks, such as image and audio processing. Modern deep learning systems can often capture this latter type of intelligence through efficient function approximation. However, deep learning has seen limited success with relational and abstract reasoning, which requires identifying novel associations from limited data and generalizing to new domains.

Recognizing the importance of this capability, machine learning research has explored several novel frameworks for relational learning (Graves et al., 2014; Pritzel et al., 2017; Santoro et al., 2017; Battaglia et al., 2018; Barrett et al., 2018; Shanahan et al., 2020; Whittington et al., 2020; Webb et al., 2021; Mondal et al., 2023). In this paper, we propose a framework that casts relational learning in terms of Transformers. The success of Transformers lies in the use of attentional mechanisms to support richly context-sensitive processing (Vaswani et al., 2017; Wolf et al., 2020; Kerg et al., 2020). However, it is clear that Transformers are missing core capabilities required for modeling human thought (Mahowald et al., 2023), including an ability to support analogy and abstraction. While large language models show a surprising ability to complete some analogies (Webb et al., 2023), this ability emerges implicitly only after processing vast amounts of data.

The Transformer architecture has the ability to model relations between objects implicitly through its attention mechanisms. However, we argue in this paper that standard attention produces entangled representations encoding a mixture of relational information and object-level features, resulting in suboptimal sample efficiency for learning relations. In this work, we propose an extension of Transformers that enables explicit relational reasoning through a novel module called the *Abstractor*. At the core of the Abstractor is a variant of attention called *relational cross-attention*. Our approach is motivated by an architectural inductive bias for relational learning we call the "relational bottleneck," which separates relational information from extraneous object-level features (see Webb et al., 2024, for a cognitive science perspective on this idea).

A growing body of literature has focused on developing machine learning architectures for relational representation learning. An early example is the Relation Network (Santoro et al., 2017), which

proposes modeling pairwise relations by applying an MLP to the concatenation of object representations. Shanahan et al. (2020) proposed the PrediNet architecture, which learns representations of relations in a manner inspired by predicate logic. Webb et al. (2021)'s ESBN is a memory-augmented LSTM network that aims to factor representations into 'sensory' and 'relational'. Another related architecture is CoRelNet (Kerg et al., 2022), which reduces relational learning to modeling a similarity matrix. More recently, Altabaa and Lafferty (2023) tackled the problem of learning representations of hierarchical relations by formalizing a notion of "relational convolution".

The Transformer is a common baseline against which other approaches are compared in this literature. These works show that explicitly relational architectures outperform the Transformer on several synthetic discriminative relational tasks, sometimes by large margins (Shanahan et al., 2020; Webb et al., 2021; Kerg et al., 2022; Altabaa and Lafferty, 2023). We offer an explanation, arguing that while the Transformer architecture is versatile enough to learn such relational tasks given enough data, it does not support relational representation explicitly and thus can suffer in terms of sample-efficiency. The Abstractor extends the Transformer framework by introducing an inductive bias for learning representations of relations that are disentangled from extraneous object-level features.

Our experiments first compare the Abstractor to other relational architectures on discriminative relational tasks, finding that the Abstractor is both more flexible and achieves superior sample efficiency. We then evaluate whether the Abstractor can augment a Transformer to improve relational reasoning by evaluating on synthetic *sequence-to-sequence* relational tasks, which has so far been unexplored in the literature on explicitly relational architectures. Finally, we evaluate an Abstractor-supported architecture on a set of mathematical problem-solving tasks to evaluate the potential of the idea on tasks more representative of real-world applications. We observe consistent, sometimes dramatic, gains in sample efficiency.

## 2 RELATIONAL CROSS-ATTENTION AND THE ABSTRACTOR MODULE

At a high level, the primary function of an Abstractor is to compute abstract relational features of its inputs.[1] That is, given a sequence of input objects $x_1, \ldots, x_n$, the Abstractor learns to model a relation $r(\cdot, \cdot)$ and computes a function on the set of pairwise relations between objects $\{r(x_i, x_j)\}_{ij}$. At the heart of the Abstractor module is an inductive bias we call the *relational bottleneck*, that disentangles relational information from the features of individual objects.

### 2.1 MODELING RELATIONS AS INNER PRODUCTS

A "relation function" maps a pair of objects $x_1, x_2 \in \mathcal{X}$ to a vector representing the relation between the two objects. We model pairwise relations as inner products between appropriately encoded (or 'filtered') object representations. In particular, we model the pairwise relation function $r(\cdot, \cdot) \in \mathbb{R}^{d_r}$ in terms of $d_r$ learnable 'query' encoders $\phi_q^1, \ldots, \phi_q^{d_r}$ and 'key' encoders $\phi_k^1, \ldots, \phi_k^{d_r}$,

$$r(x, y) = \left( \langle \phi_q^1(x), \phi_k^1(y) \rangle, \langle \phi_q^2(x), \phi_k^2(y) \rangle, \ldots, \langle \phi_q^{d_r}(x), \phi_k^{d_r}(y) \rangle \right) \in \mathbb{R}^{d_r}. \tag{1}$$

Modeling relations as inner products $\langle \phi_q(x), \phi_k(y) \rangle$ ensures that the result represents a comparison between the two objects' features. More precisely, inner product relations induce a geometry on the object space $\mathcal{X}$, with notions of distance, angles, and orthogonality. Altabaa and Lafferty (2024) analyzes the approximation properties of inner products of neural networks for relation functions.

Considering all pairwise relations yields a *relation tensor*, $R = [r(x_i, x_j)]_{i,j} \in \mathbb{R}^{n \times n \times d_r}$.

### 2.2 RELATIONAL CROSS-ATTENTION

The core operation in a Transformer is attention. For an input sequence $X = (x_1, \ldots, x_n) \in \mathbb{R}^{n \times d}$, self-attention transforms the sequence via, $X' \leftarrow \mathrm{Softmax}(\phi_q(X)\phi_k(X)^\top) \phi_v(X)$, where $\phi_q, \phi_k, \phi_v$ are functions applied independently to each object in the sequence. Note that $R :=$

---

[1] In this paper, we will use the name 'Abstractor' to refer to both the module and to model architectures which contain the Abstractor module as a main component.

$\phi_q(X)\phi_k(X)^\top$ is a relation matrix in the sense defined above. Self-attention admits an interpretation as a form of "neural message-passing" (Gilmer et al., 2017) as follows

$$x_i' \leftarrow \text{MessagePassing}\left(\{(\phi_v(x_j), R_{ij})\}_{j\in[n]}\right) = \sum_j \bar{R}_{ij}\phi_v(x_j), \tag{2}$$

where $m_{j\rightarrow i} = (\phi_v(x_j), R_{ij})$ is the message from object $j$ to object $i$, encoding the sender's features $\phi_v(x_j)$ and the relation between the two objects $R_{ij} = \langle\phi_q(x_i), \phi_k(x_j)\rangle$. Here, $\bar{R} = \text{Softmax}(R)$ is the softmax-normalized relation matrix. Hence, the processed representation obtained by self-attention is an entangled mixture of relational information and object-level features.

Our goal is to learn relational representations that are abstracted away from object-level features in order to achieve more sample-efficient learning and improved generalization in relational reasoning. This is not naturally supported by the entangled representations produced by standard self-attention. We achieve this via a simple modification of attention—we replace the values $\phi_v(x_i)$ with vectors that *identify* objects, but do not encode their features. We call those vectors *symbols*. Hence, the message sent from object $j$ to object $i$ is now $m_{j\rightarrow i} = (s_j, R_{ij})$, the relation between the two objects, together with the symbol identifying the sender object,

$$A_i \leftarrow \text{MessagePassing}\left(\{(s_j, R_{ij})\}_{j\in[n]}\right) = \sum_j \bar{R}_{ij}s_j. \tag{3}$$

Symbols act as abstract references to objects, akin to pointers in traditional symbolic architectures. They do not encode information about the contents or features of the objects, but rather *refer* to objects. This results in improved sample efficiency and generalization by restricting the search space of computations, and allowing abstraction through shared symbols. We call the vectors $\{s_i\}$ 'symbols' in the same sense that we call '$x$' a symbol in an equation like $y = x^2$—they reference an object with an unspecified value. Suppose for now that each object $x_i$ is assigned a symbol $s_i$ in a manner that satisfies this property. We will discuss symbol-assignment mechanisms in the next subsection.

This modification yields a variant of attention that we call *relational cross-attention* (RCA), given by

$$A \leftarrow \sigma_{\text{rel}}\left(\phi_q(X)\,\phi_k(X)^\top\right) S, \tag{4}$$

where $S = (s_1, \ldots, s_n)$ are the symbols, $\sigma_{\text{rel}}$ is the relation activation function, and $\phi_q, \phi_k$ correspond to the query and key transformations. When the relation activation function $\sigma_{\text{rel}}$ is softmax, this corresponds to $\text{Attention}(Q \leftarrow X, K \leftarrow X, V \leftarrow S)$. In contrast, self-attention corresponds to $\text{Attention}(Q \leftarrow X, K \leftarrow X, V \leftarrow X)$, mixing relational information with object-level features.

We observe in our experiments that allowing $\sigma_{\text{rel}}$ to be a configurable hyperparameter can lead to performance benefits in some tasks. Softmax has the effect of normalizing the relation between a pair of objects $(i, j)$ based on the strength of $i$'s relations with the other objects in the sequence. In some tasks this is useful. In other tasks, this may mask relevant information, and element-wise activations (e.g., tanh, sigmoid, or linear) may be more appropriate.

Relational cross-attention implements a type of information bottleneck, that we call the "relational bottleneck", wherein the resultant representation encodes only relational information about the object sequence (Figure 1) and does not encode information about the features of individual objects. This enables a branch of the model to focus purely on modeling the relations between objects, yielding greater sample efficiency in tasks that rely on relational reasoning.

Multi-dimensional relations can be modeled through multi-head relational cross-attention. In our experiments, $\phi_q^i, \phi_k^i$ are linear maps $W_q^i, W_k^i$, and multi-head relational cross-attention is given by

$$\text{RelationalCrossAttention}(X, S) = \text{concat}\left(A^1, \ldots, A^{d_r}\right)W_o,$$
$$\text{where } A^i = \sigma_{\text{rel}}\left((X\,W_q^i)(X\,W_k^i)^\top\right)S\,W_o^i. \tag{5}$$

## 2.3 SYMBOL ASSIGNMENT MECHANISMS

Abstraction relies on the assignment of symbols to individual objects without directly encoding their features. We propose three different mechanisms for assigning symbols to objects.

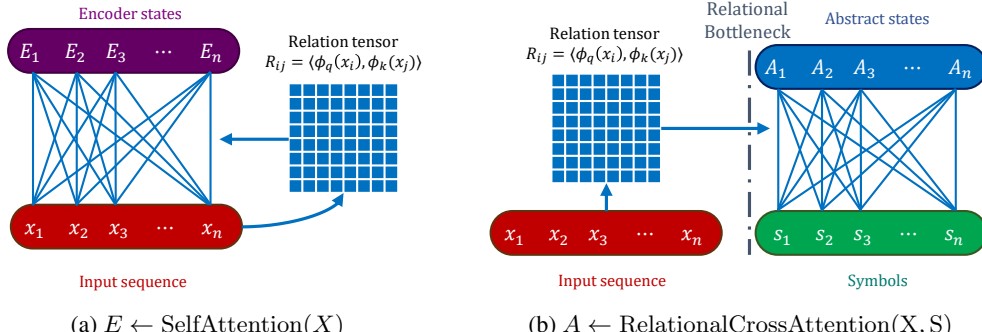

(a) $E \leftarrow \mathrm{SelfAttention}(X)$  (b) $A \leftarrow \mathrm{RelationalCrossAttention}(\mathrm{X}, \mathrm{S})$

Figure 1: Comparison of relational cross-attention with self-attention. Red represents object-level features, blue represents relational features, and purple represents mixed representations. Relational cross-attention computes relational information disentangled from the features of individual objects.

**Positional symbols.** The simplest symbol assignment mechanism is to assign symbols to objects sequentially based on the order they appear in the sequence. That is, we maintain a library of symbols $S_{\mathrm{lib}} = (s_1, \ldots, s_{\mathtt{max\_len}}) \in \mathbb{R}^{\mathtt{max\_len} \times d_{\mathrm{model}}}$, and assign the symbol $s_i$ to the $i$-th object. The symbol library $S_{\mathrm{lib}}$ can be either learned parameters of the model or fixed positional embeddings.

**Position-relative symbols.** Similar to relative positional embeddings (Shaw et al., 2018; Kazemnejad et al., 2023), we can compute relational cross-attention with position-relative symbols via $A_i \leftarrow \sum_j R_{ij} s_{j-i}$, where the symbol library is $S_{\mathrm{lib}} = (\ldots, s_{-1}, s_0, s_1, \ldots)$.

**Symbolic attention.** In this case we learn a library of symbols $S_{\mathrm{lib}} = (s_1, \ldots, s_{n_s}) \in \mathbb{R}^{n_s \times d_{\mathrm{model}}}$ together with associated "binding vectors" $B_{\mathrm{lib}} = (b_1, \ldots, b_{n_s})$, where $n_s$ is the number of symbols in the library. Through an attention mechanism, symbols are bound to objects $x_i$ based on their relations to the symbol binding vectors. A multi-head variant of symbolic attention is naturally defined by concatenating the symbols retrieved for each head. Formally,

$$\mathrm{SymbolicAttention}(X) = \mathrm{concat}\left(S^{(1)}, \ldots, S^{(n_h)}\right),$$
$$S^i = \mathrm{Softmax}\left((XW_q^i){B_{\mathrm{lib}}^i}^{\top}\right) S_{\mathrm{lib}}^i. \tag{6}$$

In terms of representational capacity, this is similar to cross-attending to the symbol parameters $S_{\mathrm{lib}}$. Note that symbolic attention weakens the relational bottleneck since object-level features are used to retrieve a symbol for each object. However, the symbols are shared across objects and sequences, and the dependence is only with respect to a low-dimensional projection of the object-level features, which we may think of as encoding the object's "syntactic role." Encoding such information in the symbols allows identifying objects by their role, rather than merely their position or relative position.

The choice of symbol assignment mechanism determines the way in which relational information is encoded in the abstract states. For example, with positional symbols, $A_i$ encodes the relations between object $i$ and each object $j$, identifying $j$ by its position (or relative position in the case of position-relative symbols). In contrast, with symbolic attention, each object $j$ is identified by its "syntactic role," as determined by its relation to the binding vectors.

## 2.4 THE ABSTRACTOR MODULE

We now describe the Abstractor module. Like the Encoder in a Transformer, this is a module that processes an input sequence of objects $X = (x_1, \ldots, x_n)$ producing a processed sequence of objects $A = (A_1, \ldots, A_n)$ that represents features of the input sequence. In an Encoder, the output objects represent a mix of object-level features and relational information. In an Abstractor, the output objects are abstract states that represent relational information, abstracted away from the features of individual objects. The core operation in an Abstractor module is relational cross-attention. Mirroring an Encoder, an Abstractor module can comprise several layers, each composed of relational cross-attention followed by a feedforward network. Optionally, residual connections

and layer-normalization can be applied as suggested by Vaswani et al. (2017)[2]. The algorithmic description is presented in Algorithm 1. In Appendix B, we theoretically study the approximation properties of the Abstractor module, showing that it is a universal approximator of relation functions.

---

**Algorithm 1:** Abstractor module

**Input :** object sequence: $\boldsymbol{X} = (x_1, \ldots, x_n) \in \mathbb{R}^{n \times d}$

$A^{(0)} \leftarrow \text{SymbolicAttention(X)}$ `  // or,` $A^{(0)} \leftarrow S_{1:n}$`, for positional symbols`
**for** $l \leftarrow 1$ **to** $L$ **do**
     $A^{(l)} \leftarrow \text{RelationalCrossAttention}\left(X, A^{(l-1)}\right)$
     $A^{(l)} \leftarrow A^{(l)} + A^{(l-1)}$ ` // residual connection (optional)`
     $A^{(l)} \leftarrow \text{LayerNorm}(A^{(l)})$ ` // (optional; can also be done pre-RCA)`
     $A^{(l)} \leftarrow \text{FeedForward}\left(A^{(l)}\right)$
**end**
**Output:** $A^{(L)}$

---

The hyperparameters of an Abstractor module include the number of layers $L$, the relation dimension $d_r$ (i.e., number of heads), the projection dimension $d_k$ (i.e., key dimension), the relation activation function $\sigma_{\text{rel}}$, and the model dimension $d_{\text{model}}$. The learnable parameters, at each layer, are the projection matrices $W_q^i, W_k^i \in \mathbb{R}^{d_{\text{model}} \times d_k}$, $i \in [d_r]$, the symbol library $S_{\text{lib}}$, and the parameters of the feedforward network. In our experiments, we use a 2-layer feedforward network with a hidden dimension $d_{\text{ff}}$ and ReLU activation. The implementation in the publicly available code adds a few additional hyperparameters, including whether to restrict the learned relations to be symmetric (via $W_q^i = W_k^i$), and whether to apply self-attention after relational cross-attention (which enables exchange of relational information between abstract states).

## 3 ABSTRACTOR ARCHITECTURES

Whereas a Transformer Encoder performs "general-purpose" processing, extracting representations of both object-level and relational information, an Abstractor module is more specialized and produces more enriched relational representations. An Abstractor module can be integrated into a broader transformer-based architecture, for enhanced relational processing.

To facilitate the discussion of different architectures, we distinguish between two types of tasks. In a *purely relational* prediction task, there exists a sufficient statistic of the input which is purely relational and encodes all the information that is relevant for predicting the target. The experiments of (Webb et al., 2021; Kerg et al., 2022) are examples of purely relational discriminative tasks. We consider discriminative relational tasks in Section 4.1. An example of a purely relational *sequence-to-sequence* task is the object-sorting task described in Section 4.2. Many real-world tasks, however, are not purely relational. In a *partially-relational* prediction task, the relational information is important but is not sufficient for predicting the target. The math problem-solving experiments in Section 4.3 are partially-relational. Natural language understanding is also an example of a partially-relational task.

The way that an Abstractor module is integrated into a broader model architecture should be informed by the underlying prediction task. Figure 2 depicts several Abstractor architectures each with different configurations. Architecture (a) depicts a configuration in which the Abstractor processes the relational features in the input, and the decoder attends to the abstract states $A$. Architecture (b) depicts a configuration in which the input objects are first processed by an Encoder, followed by an Abstractor for relational processing, and the decoder again attends to the abstract states. These architectures would be appropriate for purely relational tasks since the decoder attends only to the relational representations in the abstract states. Architectures (c) and (d), in which information can also pass directly from the encoder to the decoder, would be more appropriate for more general tasks that are only partially relational. For example, in architecture (c), the model branches into two parallel processing streams in which an Encoder performs general-purpose processing and an Abstractor

---

[2]Layer-normalization can also be applied *before* relational cross-attention, as in the pre-LN Transformer (Xiong et al., 2020).

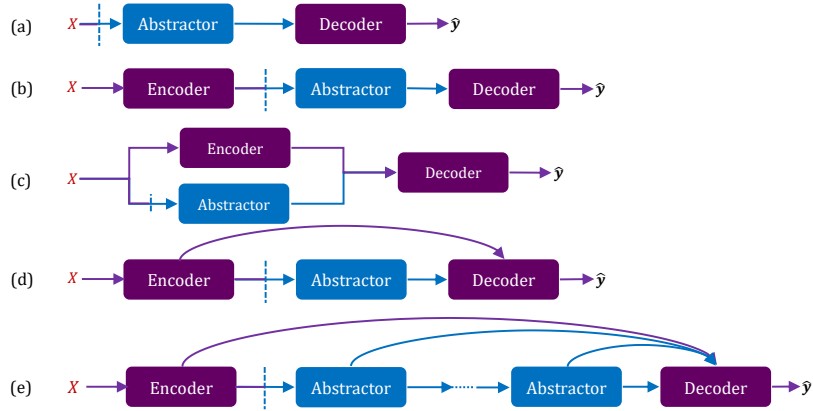

Figure 2: Examples of Abstractor-based model architectures.

performs more specialized processing of relational information. The decoder attends to *both* the encoder states and the abstract states. These architectures use the "multi-attention decoder" described in Appendix A. Finally, architecture (e) depicts a *composition* of Abstractors, wherein the abstract states produced by one Abstractor module are used as input to another Abstractor. This results in computing "higher-order" relational information (i.e., relations on relations).

## 4 EXPERIMENTS

### 4.1 DISCRIMINATIVE RELATIONAL TASKS

**Order relations: modeling asymmetric relations.** We generate $N = 64$ "random objects" represented by iid Gaussian vectors, $o_i \sim \mathcal{N}(0, I) \in \mathbb{R}^{32}$, and associate an order relation to them $o_1 \prec o_2 \prec \cdots \prec o_N$. Note that $\prec$ is *anti-symmetric*. Of the $N^2 = 4096$ possible pairs $(o_i, o_j)$, 15% are held out as a validation set (for early stopping) and 35% as a test set. We evaluate learning curves by training on the remaining 50% and computing accuracy on the test set (10 trials for each training set size). Note that the models are evaluated on pairs they have never seen. Thus, the models will need to generalize based on the transitivity of the $\prec$ relation.

We compare five models: an Abstractor, standard (symmetric) CoRelNet, an asymmetric variant of CoRelNet, PrediNet, and an MLP. The MLP, which is a non-relational architecture, is completely unable to learn the task. Among the relational architectures, we observe that standard CoRelNet is also completely unable to learn the task, whereas the Abstractor and asymmetric CoRelNet learn the transitive $\prec$ relation (Figure 4a). PrediNet has limited success in learning the task. This can be explained by the observation that symmetric inner products (e.g., in standard CoRelNet) don't have the representational capacity to model asymmetric relations, whereas the asymmetric inner products with different learned left and right encoders do.

***SET*: modeling multi-dimensional relations.** *SET* is a cognitive task in which players are presented with a sequence of cards, each of which contains figures that vary along four dimensions (color, number, pattern, and shape) and they must find triplets of cards that obey a deceptively simple rule: along each dimension, cards in a "set" must either have the same value or three unique values (Figure 3). In this experiment, the task is to classify triplets of card images as "set" or not.

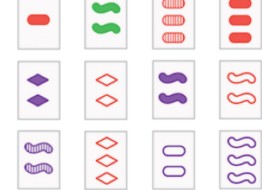

Figure 3: The *SET* game

Again, we compare an Abstractor, CoRelNet, PrediNet, and an MLP. The shared architecture is `CNN` → $\{\cdot\}$ → `Flatten` → `Dense`, where $\{\cdot\}$ is one of the aforementioned modules. The CNN embedder is obtained through a pre-training task. We report learning curves in Figure 4b (10 trials per training set size). We find that the Abstractor model significantly outperforms the other baselines. We attribute this to its relational inductive biases and its ability to model multi-dimensional relations. In this task, there exist four different relations (one for each attribute) that are needed to

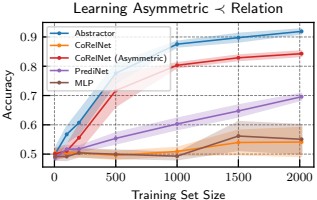 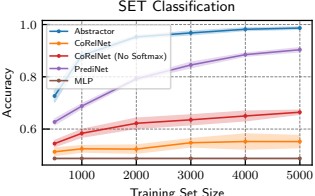 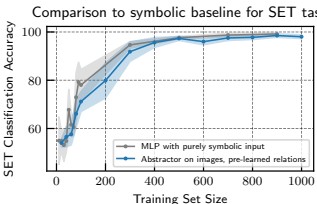

(a) The $\prec$ relation can be learned with asymmetric but not symmetric inner products.

(b) The Abstractor's ability to model multi-dimensional relations enables it to solve *SET*.

(c) Comparison of Abstractor trained on card images and MLP with hand-encoded relations.

Figure 4: Experiments on discriminative relational tasks and comparison to CoRelNet.

determine whether a triple of cards forms a set. We hypothesize that the ability to model relations as multi-dimensional is also part of the reason that the Abstractor is more sample efficient in learning the order relation in the previous experiment—even though the underlying relation is "one-dimensional", having a multi-dimensional representation enables greater robustness and multiple avenues towards a good solution during optimization.

***SET* (continued): comparison to "neuro-symbolic" model.** In Figure 4c, to evaluate the quality of the *representations* produced by Abstractors, we compare an Abstractor-based model to a "neuro-symbolic" model which receives as input a binary representation of the four relevant relations. We train 1-head Abstractors separately for each of the four attributes to learn "same/different" relations, where the task is to decide if an input pair of cards is the same or different for that attribute. We then use the $W_q$ and $W_k$ parameters learned for these relations to initialize the relations in a multi-head Abstractor. The Abstractor is then trained on a dataset of triples of cards, half of which form a "set".

This is compared to a baseline neuro-symbolic model where, instead of images, the input is a vector with 12 bits, explicitly encoding the relations. A two-layer MLP is then trained to decide if the triple forms a "set". The MLP using the symbolic representation represents an upper bound on the performance achievable by any neural network model. This comparison shows that the relational representations learned by an Abstractor result in sample efficiency that is not far from that obtained with pre-specified symbolic encodings of the relevant relations.

## 4.2 OBJECT-SORTING: PURELY RELATIONAL SEQUENCE-TO-SEQUENCE TASKS

In the following set of experiments, we consider sequence-to-sequence tasks which are purely relational, and compare an Abstractor-supported model to a standard Transformer. We consider the task of *sorting* sequences of random objects. This task is "purely relational" in the sense that there exists a relation (order) which is a sufficient statistic for solving the task—the features of individual objects beyond this relation are extraneous. This is a more controlled setting that tests the hypothesis that the inductive biases of the Abstractor confer benefits in modeling relations. The experiments in the present section demonstrate that the Abstractor enables a dramatic improvement in sample-efficiency on sequence-to-sequence relational tasks.

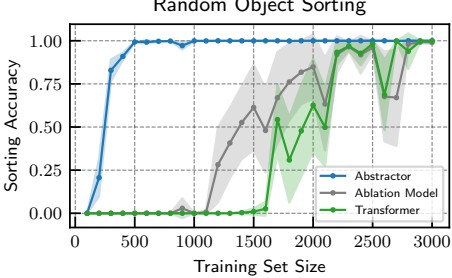 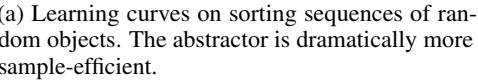 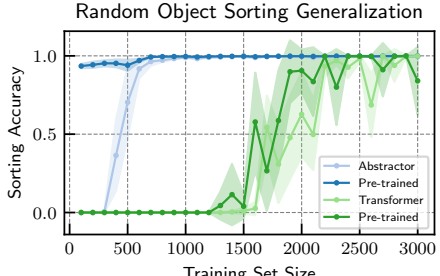

(a) Learning curves on sorting sequences of random objects. The abstractor is dramatically more sample-efficient.

(b) Learning curves with and without pre-training on a similar sorting task. The Abstractor benefits from pre-training.

Figure 5: Experiments on object-sorting, a purely relational sequence-to-sequence task.

```
Task: polynomials__expand              Task: algebra__linear_1d
Question: Expand (2*x + 3)*(x - 1).    Question: Solve for z:  5*z + 2 = 9.
Answer: 2*x**2 + x - 3                  Answer: 7/5
```

Figure 6: Examples of input/target sequences from the math problem-solving dataset.

**Superior sample-efficiency on relational seq2seq tasks.** We generate random objects in the following way. First, we generate two sets of random attributes $\mathcal{A} = \{a_1, a_2, a_3, a_4\}$, $a_i \sim \mathcal{N}(0, I) \in \mathbb{R}^4$ and $\mathcal{B} = \{b_1, \ldots, b_{12}\}$, $b_i \sim \mathcal{N}(0, I) \in \mathbb{R}^8$. To each set of attributes, we associate the strict ordering relation $a_1 \prec a_2 \prec a_3 \prec a_4$ and $b_1 \prec b_2 \prec \cdots \prec b_{12}$, respectively. Our random objects are formed by the Cartesian product of these two attributes $\mathcal{O} = \mathcal{A} \times \mathcal{B}$, yielding $N = 48$ objects. We associate with $\mathcal{O}$ the following strict ordering relation: $(a_i, b_j) \prec (a_k, b_l)$ if $a_i \prec a_k$ or if $a_i = a_k$ and $b_j \prec b_l$. Given a set of objects in $\mathcal{O}$, the task is to sort it according to $\prec$. The input sequences are randomly permuted sequences of 10 objects in $\mathcal{O}$ and the target sequences are the indices of the object sequences in sorted order (i.e., 'argsort'). The training data are sampled uniformly from the set of length-10 sequences in $\mathcal{O}$. We also generate non-overlapping validation and testing datasets.

We evaluate learning curves on an Abstractor, a standard Transformer, and an "Ablation" model (10 trials for each training set size). The Abstractor uses architecture (b) in Figure 2 with learned positional symbols. The Encoder-to-Abstractor interface uses relational cross-attention and the Abstractor-to-Decoder interface uses standard cross-attention. The Ablation Model tests the effects of relational cross-attention in the Abstractor model—it is architecturally identical to the Abstractor model with the crucial exception that the Encoder-to-Abstractor interface instead uses standard cross-attention. The hyperparameters of the models are chosen so that the parameter counts are similar (details in Appendix C). We find that the Abstractor is dramatically more sample-efficient than both the standard Transformer and the Ablation model (Figure 5a).

**Ability to generalize to similar tasks.** We also used the object-sorting task and the dataset generated as described above to test the Abstractor's ability to generalize from similar relational tasks through pre-training. The main task uses the same dataset described above. The pre-training task involves the same object set $\mathcal{O}$ but with a modified order relation. The ordering in attribute $\mathcal{A}$ is randomly permuted, while the ordering in attribute $\mathcal{B}$ is kept the same. We pre-train an Abstractor and a Transformer on the pre-training task and then, using those learned weights for initialization, evaluate learning curves on the original task. Since the Transformer requires more training samples to learn the object-sorting task, we use a pre-training set size of $3\,000$, chosen to be large enough for the Transformer to learn the pre-training task.

This experiment assesses the models' ability to generalize relations learned on one task to a new task. Figure 5b shows the learning curves for each model with and without pre-training. We observe that when the Abstractor is pre-trained, its learning curve on the object-sorting task is significantly accelerated, whereas the Transformer does not benefit from pre-training. We attribute this to the fact that the Transformer's learned representations are entangled with extraneous object-level features, which prevents generalization; by contrast, the Abstractor's disentangled relational representations can be more easily mapped to the new task.

### 4.3 Math problem-solving: partially-relational sequence-to-sequence tasks

The object-sorting experiments in the previous section are "purely relational" in the sense that the set of pairwise $\prec$ order relations is a sufficient statistic for solving the task. In a general sequence-to-sequence task, however, there may not be a relation that is a sufficient statistic. Nonetheless, relational reasoning may still be crucial for solving the task, and the enhanced relational reasoning capabilities of the Abstractor may enable performance improvements. The "partially-relational" architectures described in Section 3 enable a branch of the model to focus on relational reasoning while another branch performs general processing involving object-level attributes. In this section, we evaluate such an Abstractor model (using architecture (d) of Figure 2) on a set of math problem-solving tasks based on the dataset proposed by Saxton et al. (2019).

The dataset consists of several math problem-solving tasks, with each task having a set of question-answer pairs. The tasks include solving equations, expanding products of polynomials, differentiating functions, predicting the next term in a sequence, etc. A sample of question-answer pairs is displayed in Figure 6. The overall dataset contains $2 \times 10^6$ training examples and $10^4$ validation examples per task. Questions have a maximum length of 160 characters and answers have a maximum length

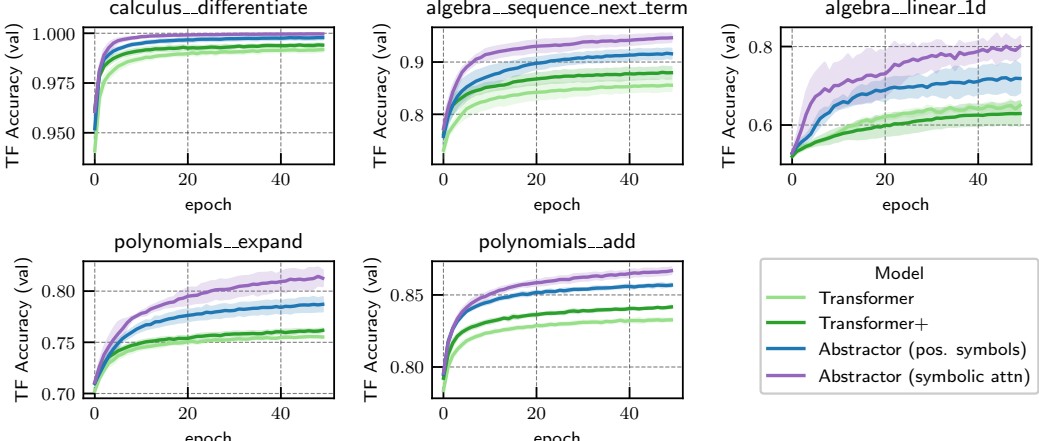

Figure 7: Training curves comparing an Abstractor-based architecture to a standard Transformer on mathematics problem-solving tasks.

of 30 characters. We use character-level encoding with a common alphabet of size 95 (including upper/lower case characters, digits, punctuation, and special start/end/pad tokens).

We compare an Abstractor model using architecture (d) in Figure 2 to a standard Transformer. We evaluate an Abstractor with positional symbols and an Abstractor with symbolic attention (see Section 2.3). Since the Abstractor-based models with architecture (d) have an Abstractor module in addition to an Encoder and Decoder, we compare against two versions of the Transformer in order to control for parameter count. In the first, the Encoder and Decoder have identical hyperparameters to the Abstractor model. In the second, we increase the model dimension and hidden layer size of the feedforward network such that the overall parameter count is approximately the same as for the Abstractor model. We refer to the first model as "Transformer" and the second as "Transformer+" in the figures. The precise architectural details and hyperparameters are described in Appendix C.

We evaluate the three models on five subtasks: differentiating functions (calculus); predicting the next term in a sequence (algebra); solving a linear equation (algebra); expanding polynomials; and adding polynomials. For each, we train on the training split and track the teacher-forcing accuracy (excluding null characters) on the validation split. For each combination of model and task, we repeat the experiment five times and report error bars as twice the standard error of the mean.

Figure 7 shows the validation teacher-forcing accuracy during the course of training, which we use as a proxy for sample efficiency. We observe an improvement in accuracy compared to both 'Transformer' and 'Transformer+' across all tasks. The larger Transformer tends to perform better than the smaller Transformer, but the Abstractor-based model consistently outperforms both, with the symbolic attention mechanism showing the greatest improvement. This indicates that the performance improvement stems from the architectural modification and inductive bias. We conjecture that a "partially-relational" Abstractor architecture (e.g., architecture (d)) implements two branches of information processing. The Encoder performs more general-purpose processing of the input sequence, while the Abstractor performs more specialized relational processing. The Decoder then has access to both representations, enabling it to perform the task more effectively.

## 5 CONCLUSION

In this work, we propose a variant of attention that produces representations of relational information disentangled from object-level attributes. This leads to the development of the Abstractor module, which fits naturally into the powerful framework of Transformers. Through a series of experiments, we demonstrate the potential of this new framework to achieve gains in sample efficiency in both purely relational tasks and more general sequence modeling tasks. This work opens up several avenues for future research, including a better understanding of the strengths and weaknesses of different architectural variants, work towards a more streamlined scalable architecture, and exploring the framework's use in increasingly complex real-world problems.

CODE AND REPRODUCIBILITY

Code, detailed experimental logs, and instructions for reproducing our experimental results are available at: https://github.com/Awni00/Abstractor.

ACKNOWLEDGMENTS

This research was supported by the funds provided by the National Science Foundation and by DoD OUSD (R&E) under Cooperative Agreement PHY-2229929 (The NSF AI Institute for Artificial and Natural Intelligence). JL was supported in part by NSF grant CCF-1839308 and NSF grant DMS-2015397. JDC was supported by a Vannevar Bush Faculty Fellowship.

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

## A MULTI-ATTENTION DECODER

---

**Algorithm 2:** Multi-Attention Decoder

---

**Input :** Target sequence: $\boldsymbol{y} = (y_0, \ldots, y_{l_y-1})$,

Context sequences: $X^{(i)} = (x_1^{(i)}, \ldots, x_{l_i}^{(i)}), \ i = 1, \ldots, K$

$D^{(0)} \leftarrow \boldsymbol{y}$
**for** $l \leftarrow 1$ **to** $L$ **do**
    $D^{(l)} \leftarrow \mathrm{CausalSelfAttention}\left(D^{(l-1)}\right)$
    `residual connection and layer-normalization`
    **for** $i \leftarrow 1$ **to** $K$ **do**
        $D^{(l)} \leftarrow \mathrm{CrossAttention}\left(D^{(l)}, X^{(i)}\right)$
        `residual connection and layer-normalization`
    **end**
    $D^{(l)} \leftarrow \mathrm{FeedForward}\left(D^{(l)}\right)$
**end**
**Output:** $D^{(L)}$

---

## B UNIVERSAL APPROXIMATION OF RELATION FUNCTIONS

In this section, we characterize the function class of the Abstractor and relational cross-attention. We will show that, in an appropriate sense, a single Abstractor layer can compute a sequence of abstract states $A = (A_1, \ldots, A_n)$ such that $A_i$ approximates an arbitrary function of object $i$'s relations with the other objects in the input. Recall that relational cross-attention takes the form

$$A \leftarrow \sigma_{\mathrm{rel}}\left(\phi_q(X)\phi_k(X)^\top\right) S,$$

where $\phi_q, \phi_k : \mathcal{X} \to \mathbb{R}^{d_{\mathrm{proj}}}$ are multi-layer perceptrons, and $S$ is the matrix of symbols. In this analysis, we consider $\sigma_{\mathrm{rel}} : x \mapsto x$ to be the linear activation function. Further, we consider positional symbols. For simplicity, we will assume the single-head variant of relational cross-attention. The function class results derived here would of course carry over to the multi-head case, where each 'head' can approximate a function in the function class.

Recall that an Abstractor module comprises several layers, each composed of relational cross-attention and a feedforward layer. Hence, the overall operation in one layer of an Abstractor module is

$$\mathrm{AbstractorLayer}(X) = \mathrm{FeedForward}\left(\phi_q(X)\phi_k(X)^\top S\right). \tag{7}$$

We will characterize the function class $\mathrm{AbstractorLayer} : \mathcal{X}^n \to \mathbb{R}^{n \times d}$ induced by varying the parameters of $\phi_q$, $\phi_k$, FeedForward and $S$.

**Remark 1.** *In Equation (4), we formulate relational cross-attention with the maps $\phi_q, \phi_k$ as linear projections, whereas $\phi_q, \phi_k$ are multi-layer perceptrons in Equation (7). However, recall that the input to each Abstractor layer is the output of the preceding layer which ends with a multi-layer perceptron. The function class of a multi-layer perceptron followed by two different linear projections is the same as the function class of two different multi-layer perceptrons. We focus on Equation (7) for ease of presentation.*

The following result shows that a single-layer abstractor returns a sequence of abstract states which can approximate an arbitrary function of each object's relations with the other object in the input. The result is based on the analysis in (Altabaa and Lafferty, 2024) which characterizes the function class of inner products of neural networks.

**Theorem 1.** *Suppose $\mathcal{X}$ is a compact Euclidean space. Let $r : \mathcal{X} \times \mathcal{X} \to \mathbb{R}$ be any continuous relation function, and $f : \mathbb{R}^n \to \mathbb{R}^d$ any continuous function. Consider the function $g : \mathcal{X}^n \to \mathbb{R}^{n \times d}$ defined by*

$$(x_1, \ldots, x_n) \mapsto (f(R_1), \ldots, f(R_n)),$$

*where $R_i = (r(x_i, x_j))_{j \in [n]}$ is the vector of $x_i$'s relations with the other objects in the input, according to $r$. Then, for any $\varepsilon > 0$, there exists MLPs $\phi_q, \phi_k$, FeedForward and a choice of symbols $S$ such that the Abstractor layer approximates $g$ in the sup-norm*

$$\|g(x_1, \ldots, x_n) - \text{AbstractorLayer}(x_1, \ldots, x_n)\|_\infty \leq \varepsilon, \quad \textit{Lebesgue almost-everywhere.}$$

*Proof.* Let $(A_1, \ldots, A_n) = \text{AbstractorLayer}(x_1, \ldots, x_n)$ be the abstract states returned by the abstractor layer. For each $i \in [n]$, the abstract state $A_i$ takes the form,

$$A_i = \text{FeedForward}\left(\sum_{j=1}^n \langle \phi_q(x_i), \phi_k(x_j)\rangle s_j\right).$$

where $S = (s_1, \ldots, s_n)$ are the symbols assigned to each object. Let the symbol dimension be $n$ and let $s_i = e_i$, the canonical basis vectors. Then, $A_i = \text{FeedForward}(R_i)$, where $R_i = (\langle \phi_q(x_i), \phi_k(x_j)\rangle)_{j \in [n]} \in \mathbb{R}^n$. By (Altabaa and Lafferty, 2024, Theorem 3.1), there exists MLPs $\phi_q, \phi_k$ such that their inner product approximates any continuous relation function. In particular, for any $\varepsilon_1 > 0$, there exists $\phi_q, \phi_k$ such that

$$|r(x, y) - \langle \phi_q(x), \phi_k(y)\rangle| \leq \varepsilon_1, \quad \text{Lebesgue almost-every } x, y \in \mathcal{X} \tag{8}$$

Let $\phi_q, \phi_k$ be such MLPs where $\varepsilon_1$ is to be determined later. Note that (Altabaa and Lafferty, 2024) gives a bound on the number of neurons needed in terms of the continuity of $r$ and the dimension of $\mathcal{X}$. Similarly, by the universal approximation property of MLPs (e.g., Cybenko, 1989), $f : \mathbb{R}^n \to \mathbb{R}^d$ can be approximated by FeedForward uniformly in the sup-norm. That is, for any $\varepsilon_2$ there exists FeedForward such that

$$\sup_{z \in \mathbb{R}^n} \|f(z) - \text{FeedForward}(z)\|_\infty \leq \varepsilon_2. \tag{9}$$

Let $[R]_{ij} = r(x_i, x_j)$ and $[\hat{R}]_{ij} = \langle \phi_q(x_i), \phi_k(x_j)\rangle$. Then, the difference $g(x_1, \ldots, x_n) - \text{AbstractorLayer}(x_1, \ldots, x_n)$ is given by

$$\left[f(R_{1\cdot}), \ldots, f(R_{1\cdot})\right] - \left[\text{FeedForward}(\hat{R}_{1\cdot}), \ldots, \text{FeedForward}(\hat{R}_{n\cdot})\right] \tag{10}$$

Note that $\hat{R}_{i\cdot}$ is close to $R_{i\cdot}$ by Equation (8)

$$\left\|\hat{R}_{i\cdot} - R_{i\cdot}\right\|_\infty = \max_{j \in [n]} |\langle \phi_q(x_i), \phi_k(x_j)\rangle - r(x_i, x_j)|$$
$$\leq \varepsilon_1 \quad \text{Lebesgue almost-everywhere.} \tag{11}$$

Now consider the $(i, j)$-th element of the difference in Equation (10)

$$\left|\text{FeedForward}_i(\hat{R}_{j\cdot}) - f_i(R_{j\cdot})\right| \leq \left|\text{FeedForward}_i(\hat{R}_{j\cdot}) - f_i(\hat{R}_{j\cdot})\right| + \left|f_i(\hat{R}_{j\cdot}) - f_i(R_{j\cdot})\right|$$

The first term is bounded by $\varepsilon_2$ due to Equation (9). Let $\varepsilon_2 = \varepsilon/2$. Recall that $f : \mathbb{R}^n \to \mathbb{R}^d$ is continuous, and hence for all $\epsilon > 0$ there exists $\delta_f(\varepsilon) > 0$ such that $\|z_1 - z_2\|_\infty \leq \delta(\varepsilon) \implies \|f(z_1) - f(z_2)\|_\infty \leq \varepsilon$. Letting $\varepsilon_1 = \delta(\varepsilon/2)$, implies that the second term is bounded by $\varepsilon/2$ Lebesgue almost-everywhere due to Equation (11).

This holds for all $i, j$, which completes the proof.

$\square$

## C  EXPERIMENTAL DETAILS

In this section, we give further experimental details including architectures, hyperparameters, and implementation details. All models and experiments are implemented in Tensorflow. The code is publicly available on the project repo along with detailed experimental logs and instructions for reproducing our results.

## C.1 DISCRIMINATIVE TASKS (SECTION 4.1)

### C.1.1 PAIRWISE ORDER

Each model in this experiment has the following form `input` $\to \{\cdot\} \to$ `flatten` $\to$ `MLP`, where $\{\cdot\}$ is one of the modules below and `MLP` is an MLP composed of one hidden layer with 32 neurons and ReLU activation.

**Abstractor architecture.** The Abstractor module used the following hyperparameters: number of layers $L = 1$, relation dimension $d_r = 4$, symbol dimension $d_s = 64$, projection (key) dimension $d_k = 16$, feedforward hidden dimension $d_{\text{ff}} = 64$, relation activation function $\sigma_{\text{rel}} = \text{sigmoid}$. No layer normalization or residual connection. We use positional symbols as the symbol assignment mechanism, which are learned parameters of the model. The output of the Abstractor module is flattened and passed to the MLP.

**CoRelNet architecture.** CoRelNet has no hyperparameters. Given a sequence of objects, $X = (x_1, \ldots, x_n)$, standard CoRelNet (Kerg et al., 2022) simply computes the inner product and takes the Softmax. We also add a learnable linear map, $W \in \mathbb{R}^{d \times d}$. Hence, $\bar{R} = \text{Softmax}(R), R = [\langle W x_i, W x_j \rangle]_{ij}$. The CoRelNet architecture flattens $\bar{R}$ and passes it to an MLP to produce the output. The asymmetric variant of CoRelNet is given by $\bar{R} = \text{Softmax}(R), R = [\langle W_1 x_i, W_2 x_j \rangle]_{ij}$, where $W_1, W_2 \in \mathbb{R}^{d \times d}$ are learnable matrices.

**PrediNet architecture.** We based our implementation of PrediNet (Shanahan et al., 2020) on the authors' publicly available code. We used the following hyperparameters: using 4 heads, and 16 relations, a key dimension of 4 (see the original paper for the meaning of these hyperparameters). The output of the PrediNet module is flattened and passed to the MLP.

**MLP.** The embeddings of the objects are concatenated and passed directly to an MLP. The MLP has two hidden layers each with 32 neurons and a ReLU activation.

**Training/Evaluation.** We use the crossentropy loss and the Adam optimizer with a learning rate of $10^{-2}$, $\beta_1 = 0.9, \beta_2 = 0.999, \varepsilon = 10^{-7}$. We use a batch size of 64. We train for 100 epochs and restore the best model according to validation loss. We evaluate on the test set.

### C.1.2 *SET*

The card images are RGB images of dimension $70 \times 50 \times 3$. A CNN embedder processes the images separately and produces embeddings of dimension $d = 64$ for each card. The CNN is trained to predict the four attributes of each card and then an embedding for each card is obtained from an intermediate layer (i.e., the parameters of the CNN are then frozen). Recall that the common architecture is `CNN Embedder` $\to \{\cdot\} \to$ `Flatten` $\to$ `Dense(2)`, where $\{\cdot\}$ is an Abstractor, CoRelNet, PrediNet, or an MLP. We tested against the standard version of CoRelNet, but found that it did not learn anything. We iterated over the hyperparameters and architecture to improve its performance. We found that removing the softmax activation in CoRelNet improved performance a bit. We describe hyperparameters below.

**Common embedder's architecture** The architecture is given by `Conv2D` $\to$ `MaxPool2D` $\to$ `Conv2D` $\to$ `MaxPool2D` $\to$ `Flatten` $\to$ `Dense(64, 'relu')` $\to$ `Dense(64, 'relu')` $\to$ `Dense(2)`. The embedding is extracted from the penultimate layer. The CNN is trained to predict the four attributes of each card until it reaches perfect accuracy and near-zero loss.

**Abstractor architecture** The Abstractor module has hyperparameters: number of layers $L = 1$, relation dimension $d_r = 4$, symmetric relations (i.e., $W_q^i = W_k^i, i \in [d_r]$), linear relation activation (i.e., $\sigma_{\text{rel}} : x \mapsto x$), symbol dimension $d_s = 64$, projection (key) dimension $d_k = 16$, feedforward hidden dimension $d_{\text{ff}} = 128$, and no layer normalization or residual connection. We use positional symbols as the symbol assignment mechanism, which are learned parameters of the model.

**CoRelNet architecture** Standard CoRelNet is described above. It simply computes, $R = \text{Softmax}(A), A = [\langle W x_i, W x_j \rangle]_{ij}$. This variant was stuck at 50% accuracy regardless of training set size. We found that removing the Softmax helped. Figure 4b compares against both variants of CoRelNet.

This finding suggests that allowing $\sigma_{\mathrm{rel}}$ to be a configurable hyperparameter is a useful feature of the Abstractor. Softmax performs contextual normalization of relations, such that the relation between $i$ and $j$ is normalized in terms of $i$'s relations with all other objects. This may be useful at times, but may also cripple a relational model when it is more useful to represent an absolute relation between a pair of objects, independently of the relations with other objects.

**PrediNet architecture.** We used the following hyperparameters: using 4 heads, and 16 relations, a key dimension of 4 (see the original paper for the meaning of these hyperparameters). The output of the PrediNet module is flattened and passed to the MLP.

**MLP.** The embeddings of the objects are concatenated and passed directly to an MLP. The MLP has two hidden layers each with 32 neurons and a ReLU activation.

**Data generation** The data is generated by randomly sampling a "set" with probability 1/2 and a non-"set" with probability 1/2. The triplet of cards is then randomly shuffled.

**Training/Evaluation** We use the crossentropy loss and the Adam optimizer with a learning rate of $10^{-3}$, $\beta_1 = 0.9, \beta_2 = 0.999, \varepsilon = 10^{-7}$. We use a batch size of 64. We train for 200 epochs and restore the best model according to validation loss. We evaluate on the test set.

## C.2 RELATIONAL SEQUENCE-TO-SEQUENCE TASKS (SECTION 4.2)

### C.2.1 SAMPLE-EFFICIENCY IN RELATIONAL SEQ2SEQ TASKS

**Abstractor architecture** The Abstractor model uses architecture (b) of Figure 2. For each of the Encoder, Abstractor, and Decoder modules, we use $L = 2$ layers, 2 attention heads/relation dimensions, a feedforward network with $d_{\mathrm{ff}} = 64$ hidden units and a model/symbol dimension of $d_{\mathrm{model}} = 64$. The relation activation function is $\sigma_{\mathrm{rel}} = \mathrm{Softmax}$. We use positional symbols as the symbol assignment mechanism, which are learned parameters of the model. The number of trainable parameters is $386,954$.

**Transformer architecture** We implement the standard Transformer of (Vaswani et al., 2017). For both the Encoder and Decoder modules, use matching hyperparameters per-layer but increase the number of layers. We use 4 layers, 2 attention heads, a feedforward network with 64 hidden units and a model dimension of 64. The number of trainable parameters is $469,898$. We increased the number of layers compared to the Abstractor in order to make it a comparable size in terms of parameter count.

**Ablation model architecture** The Ablation model uses an identical architecture to the Abstractor, except that the relational cross-attention is replaced with standard cross-attention at the Encoder-Abstractor interface (with $Q \leftarrow A, K \leftarrow E, V \leftarrow E$). It has the same number of parameters as the Abstractor-based model.

**Training/Evaluation** We use the crossentropy loss and the Adam optimizer with a learning rate of $10^{-3}$, $\beta_1 = 0.9, \beta_2 = 0.999, \varepsilon = 10^{-7}$. We use a batch size of 512. We train for 100 epochs and restore the best model according to validation loss. We evaluate learning curves by varying the training set size and sampling a random subset of the data at that size. Learning curves are evaluated starting at 100 samples up to 3000 samples in increments of 100 samples. Each 'sample' is a pair of input-output sequences. For each model and training set size, we evaluate 10 runs with different random seeds and report the mean and standard error of the mean.

### C.2.2 GENERALIZATION TO NEW OBJECT-SORTING TASKS

**Abstractor architecture** The Abstractor model uses architecture (a) of Figure 2. The Abstractor module uses learned positional symbols, has $L = 1$ layer, a model dimension of $d_{\mathrm{model}} = 64$, a relation dimension of $d_r = 4$, a softmax relation activation $\sigma_{\mathrm{rel}} = \mathrm{Softmax}$, and a feedforward network with $d_{\mathrm{ff}} = 64$. The decoder also has 1 layer with 4-head MHA and a 64-unit feedforward network.

**Transformer architecture** The Transformer is identical to the previous section.

**Training/Evaluation** The loss, optimizer, batch size, and learning curve evaluation steps are identical to the previous sections. Two object-sorting datasets are created based on an "attribute-product

structure"—a primary dataset and a pre-training dataset. As described in Section 4.2, the pre-training dataset uses the same random objects as the primary dataset but with the order relation of the primary attribute reshuffled. The models are trained on 3,000 labeled sequences of the pre-training task and the weights are used to initialize training on the primary task. Learning curves are evaluated with and without pre-training for each model.

### C.3   Math Problem-Solving (Section 4.3)

**Abstractor architectures.** The Abstractor models use architecture (d) of Figure 2. The Encoder, Abstractor, and Decoder modules share the same hyperparameters: number of layers $L = 1$, relation dimension/number of heads $d_r = n_h = 4$, symbol dimension/model dimension $d_s = d_{\mathrm{model}} = 128$, projection (key) dimension $d_k = 32$, feedforward hidden dimension $d_{\mathrm{ff}} = 256$. In the Abstractor, the relation activation function is $\sigma_{\mathrm{rel}} = \mathrm{softmax}$. In one model, positional symbols are used, with sinusoidal embeddings. In the other model, symbolic attention is used with a symbol library of $n_s = 256$ learned symbols, and 4-head symbolic attention.

**Transformer architecture.** The Transformer Encoder and Decoder have identical hyperparameters to the Encoder and Decoder of the Abstractor architecture.

**Transformer+ architecture.** In 'Transformer+', the model dimension is increased to $d_{\mathrm{model}} = 200$ and the feedforward hidden dimension is increased to $d_{\mathrm{ff}} = 400$. The remaining hyperparameters are the same.

**Training/Evaluation** We train each model for 50 epochs with the categorical cross-entropy loss and the Adam optimizer using a learning rate of $6 \times 10^{-4}$, $\beta_1 = 0.9, \beta_2 = 0.995, \varepsilon = 10^{-9}$. We use a batch size of 128.

### C.4   Additional Experiment: Robustness and Out-of-Distribution Generalization in the Object-Sorting Experiments

This experiment explores the Abstractor's robustness to noise and out-of-distribution generalization as compared to a standard Transformer. We consider the models in Section 4.2 and the corresponding object-sorting task. We train each model on this task using 3,000 labeled sequences. We chose the fixed training set size of 3,000 because is large enough that both the Abstractor and Transformer are able to learn the task. Then, we corrupt the objects with noise and evaluate performance on sequences in the hold-out test set where objects are replaced by their corrupted versions. We evaluate robustness to a random linear map as well as to additive noise, while varying the noise level. We evaluate over several trials, averaging over the realizations of the random noise.

On the hold out test set, we corrupt the object representations by applying a random linear transformation. In particular, we randomly sample a random matrix the entries of which are iid zero-mean Gaussian with variance $\sigma^2$, $\Phi \in \mathbb{R}^{d \times d}, \Phi_{ij} \sim \mathcal{N}(0, \sigma^2)$. Each object in $\mathcal{O}$ is then corrupted by this random linear transformation, $\hat{o}_i = \Phi o_i$, for each $i \in [48]$. We also test robustness to additive noise via $\hat{o}_i = o_i + \varepsilon_i, \varepsilon_i \sim \mathcal{N}(0, \sigma^2 I_d)$.

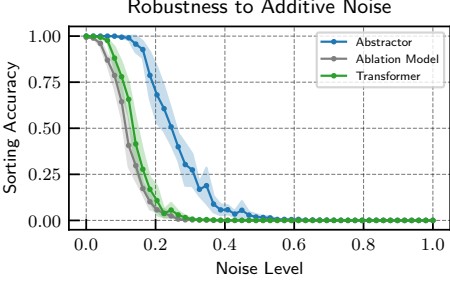
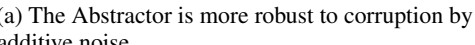
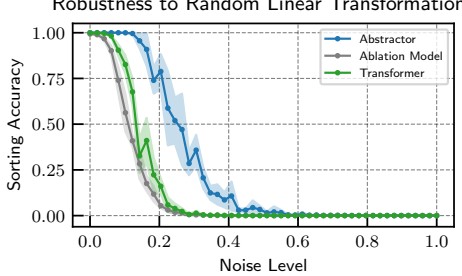

(a) The Abstractor is more robust to corruption by additive noise.

(b) The Abstractor is more robust to corruption by a random linear transformation.

Figure 8: Experiments on robustness.

The models are evaluated on the hold-out test set with objects replaced by their corrupted version. We evaluate the sorting accuracy of each model while varying the noise level $\sigma$ (5 trials at each noise level). The results are shown in figures 8a and 8b. We emphasize that the models are trained only on the original objects in $\mathcal{O}$, and are not trained on objects corrupted by any kind of noise.

This experiment can be interpreted in two lights: the first is robustness to noise. The second is a form of out-of-distribution generalization. Note that the objects seen by the models post-corruption lie in a different space than those seen during training. Hence the models need to learn relations that are in some sense independent of the value representation. As a theoretical justification for this behavior, Zhou et al., 2009 shows that $\langle \Phi x, \Phi y \rangle \approx \langle x, y \rangle$ in high dimensions, for a random matrix $\Phi$ with iid Gaussian entries. This indicates that models whose primary computations are performed via inner products, like Abstractors, may be more robust to this kind of corruption.

