# OpenReview forum: "Abstractors and relational cross-attention: An inductive bias for explicit relational reasoning in Transformers"
_ICLR.cc/2024/Conference — ICLR 2024 poster_

### Official Review · Reviewer_qTc8 · 2023-10-23

**Soundness:** 2 fair
**Presentation:** 2 fair
**Contribution:** 3 good
**Rating:** 6
**Confidence:** 3

**Summary:**

This paper presents a modification on attention that replaces the value part of self-attention with an "relational-only" representation S. This representation serves as an "information bottleneck" for the model so that it can learn separate encodings for object relations and values (eg in a separate encoder). The experimental results are strong and offer good empirical support for the effectiveness of the Abstractor layer.

**Strengths:**

- The proposed abstractor method outperforms all baselines and is more sample efficient in learning relational tasks.
- The proposed method is interesting and a relatively simple modification on top of self-attention. However, I have some questions about how S works (below)

**Weaknesses:**

- It is not clear to me how S works. Section 2.2 says "The symbols S can be either learned parameters of the model or nonparametric positional embeddings." How is S different from a positional embedding then? If S is unique per token, even if the token is a repeat of another token value-wise, then S is essentially a positional embedding. If S is unique per *value* of a token (ie all tokens of value v have the same s_i), then doesn't S implicity encode the value of the token?
- If the outputs of an abstractor layer are "abstract states that represent purely relational information" (section 2.3), then how are features associated with objects passed along/learned in models that use abstractor layers? In all the examples presented in Figure 2, abstractors are used in conjunction with regular encoders. How can you ensure that the abstractor is learning meaningful information and that all the "information flow" does not happen through the encoder layers in parlalle setups (architectures c, d, and e)?
- Section 4.1 claims that the authors hypothesize that "the ability to model relations as multi-dimensional is also the reason that the Abstractor can learn the order relation better than ..." Are there any experiments to test or verify this hypothesis? The paper also makes many claims about an information bottleneck in S, but there is no analysis on what is actually learned in S.
- Is there an ablation on model size? The baseline transformer is not very large, but it may be that a smaller transformer can learn with fewer training data points, or a larger transformer may converge faster.
- The experiments seem to be on simple problems with small models. Simple problems may not necessarily be an issue since they are relatively diverse problems, but it would be nice to see larger, more complicated problems. For example, the partial order task training set is small enough that one could consider using in context learning with a large LLM, which may offer comparable performance.

**Questions:**

See above

---

> ### Author Response · Authors · 2023-11-22
>
> > It is not clear to me how S works. Section 2.2 says "The symbols S can be either learned parameters of the model or nonparametric positional embeddings." How is S different from a positional embedding then? If S is unique per token, even if the token is a repeat of another token value-wise, then S is essentially a positional embedding. If S is unique per value of a token (ie all tokens of value v have the same s_i), then doesn't S implicity encode the value of the token?
>
> Please see the updated paper for an expanded discussion which is relevant to this question. In particular, please see section 2.3.
>
> In terms of implementation, "positional symbols" are indeed positional embeddings, which can be learned parameters of the model or fixed positional embeddings. Hence, the symbol assigned to each object does not encode its features, but identifies it (in this case, via its position). In the case of "position-relative" symbols, the identifier of a symbol encodes its relative position with respect to the 'receiver'. In the case of "symbolic attention," a symbol is retrieved from a library of learned symbols (which are parameters of the model) via attention. This allows the symbol retrieved for each object to encode something about its role, hence identifying it not only by its position, but by its role.
>
> The key inductive bias for all these symbol-assignment mechanisms is that the features of the input objects don't propagate through the values $V$ in relational cross-attention. I.e., instead of the self-attention configuration $(Q \gets X, K \gets X, V \gets X)$, in relational cross-attention we have $(Q \gets X, K \gets X, V \gets S)$, with $S$ identifying objects without encoding their features. We think that the message-passing interpretation of attention is useful to get some intuition about this.
>
> Self attention can be thought of as a form of message-passing where the message from object $j$ to object $i$ encodes the sender's features as well as the relations between the two objects, $m_{j \to i} = (\phi_v(x_j), r(x_i, x_j))$, where $r(x_i, x_j) = \langle \phi_q(x_i), \phi_k(x_j)\rangle$ is an inner product relation. Then,
> $$E_i = \mathrm{MessagePassing}(\{m_{j \to i}\}_j).$$
>
> (To make this more concrete, one could define a commutative monoid with the aggregation operation $\oplus$ as $(x_1, r_1) \oplus (x_2, r_2) = \left(\frac{\exp(r_1) x_1 + \exp(r_2) x_2}{\exp(r_1) + \exp(r_2)}, \exp(r_1) + \exp(r_2)\right)$ to formalize self-attention in the standard Message Passing Neural Networks equations. But those details are not important to the intuition.)
>
> Hence, the 'messages' in self-attention entangle relational information $r(x_i, x_j)$ with object-level information $\phi_v(x_j)$. The aim of the Abstractor is to produce disentangled relational representations, within the powerful framework of Transformers. The key idea is to replace $\phi_v(x_j)$ in $m_{j \to i}$ with vectors identifying the sender $j$ but not encoding any information about its object-level features---we call these vectors 'symbols'. The simplest way to do this is to simply sequentially assign each object in the input sequence a unique symbol, based on the position it appears (i.e., "positional symbols"). Although very simple, this works well in practice. We also explored symbolic attention as a symbol-assignment mechanism in our updated experiments (see section 4.3).

---

> ### Author Response · Authors · 2023-11-22
>
> > If the outputs of an abstractor layer are "abstract states that represent purely relational information" (section 2.3), then how are features associated with objects passed along/learned in models that use abstractor layers? In all the examples presented in Figure 2, abstractors are used in conjunction with regular encoders. How can you ensure that the abstractor is learning meaningful information and that all the "information flow" does not happen through the encoder layers in parlalle setups (architectures c, d, and e)?
>
> Thanks for this question. A minor correction, architecture (a) of Figure 2 depicts a model without a standard Encoder. The "Ability to generalize to similar tasks" experiment of section 4.2 uses architecture (a). In the discriminative relational tasks of section 4.1, the Abstractor models also do not include an Encoder (this is an Abstractor-only model with no decoder either, since these are classification tasks).
>
> But indeed, many architectures we consider do include an Encoder. The reason for this is that most real-world tasks rely on relational reasoning as well as more general-purpose sequence modeling which needs object-level features. In the architectures proposed in Section 3 and Figure 2 (e.g., architectures c, d, e), we think of the Abstractor as performing specialized relational processing in a branch of the model, while an Encoder performs more general-purpose processing in another branch. The Decoder then sequentially attends to the outputs of both.
>
> > then how are features associated with objects passed along/learned in models that use abstractor layers?
>
> This is done by the (multi-attention) Decoder when it integrates the information from the Abstractor and the Encoder.
>
> > How can you ensure that the abstractor is learning meaningful information and that all the "information flow" does not happen through the encoder layers in parlalle setups
>
> This is indicated by the performance difference compared to models which do not contain an Abstractor. For example, in the experiments of section 4.2, we compare to a standard Transformer of similar size as well as to an ablation model with matching architecture but with standard cross-attention rather than relational cross-attention. We observe a dramatic difference in performance. In the experiments of section 4.3, we compare to two Transformers of different size (one matching the hyperparameters of the Encoder/Decoder and a larger one to match parameter count). We observe a consistent difference in performance here as well.
>
> Through comparison to these controlled baselines, we can attribute the performance difference to the addition of an Abstractor module. If the Encoder was doing all the work, the Abstractor-based models would not perform better.
>
> ---------
> > Is there an ablation on model size? The baseline transformer is not very large, but it may be that a smaller transformer can learn with fewer training data points, or a larger transformer may converge faster.
>
> Please see the answer to the previous question. We did include controls to test for the following question: given a certain budget of model size (as measured by parameter count), which would yield a greater performance, incorporating an Abstractor or simply enlarging a standard Transformer? We find that incorporating an Abstractor tends to yield greater performance benefits.
>
> This was for a fixed model size, however. It would be interesting to see whether these performance benefits are persistent across model size and task complexity. We leave this to future work.
>
> ---------
> > The experiments seem to be on simple problems with small models. Simple problems may not necessarily be an issue since they are relatively diverse problems, but it would be nice to see larger, more complicated problems. For example, the partial order task training set is small enough that one could consider using in context learning with a large LLM, which may offer comparable performance.
>
> We agree that it would be interesting and important to evaluate Abstractors on larger, more complex problems. For now, this is outside the scope of this initial paper which aims to propose the idea and perform controlled evaluations.
>
> ---
>
> We hope this answers your questions and addresses some of your concerns. Please let us know if you have any further questions or comments.

---

> ### Comment · Reviewer_qTc8 · 2023-11-22
>
> Thank you for your response, I have a better understanding of your paper now. I am willing to raise my score to a 6.

---

### Official Review · Reviewer_4fms · 2023-10-30

**Soundness:** 3 good
**Presentation:** 3 good
**Contribution:** 2 fair
**Rating:** 6
**Confidence:** 3

**Summary:**

The motivation of this paper lies in addressing the challenge of relational and abstract reasoning. Contemporary deep learning models excel at tasks involving semantic and procedural knowledge, but their abilities to infer relations, draw analogies, and generalize from limited data to novel situations remain limited. The popular Transformer architecture, through its attention mechanisms, has the capacity to model relations between objects implicitly. However, these standard attention mechanisms often create entangled representations, blending relational information and object-level features in a way that is not optimal for efficient learning of relations.

This paper proposes a new variant of Transformers, named the Abstractor. The core of the Abstractor is a variant of attention, named relational cross-attention. Compared to the vanilla attention in the Transformer, relational cross-attention disentangles the object-level feature and relational information by replacing the value vector in vanilla attention with a learnable symbol vector or positional function, which is independent of the object-level feature (X). The authors demonstrate the effectiveness of their Abstractor through "pure relational" and "partial relational" tasks, and compare the Abstractor to previous architectures designed for relational tasks.

**Strengths:**

1. The overall problem is interesting. As the author has noted, human brains can perform tasks involving analogy and abstraction with limited experience, whereas current models require vast amounts of data to acquire such abilities.

2. The proposed Abstractor, a novel variant of the Transformer model, disentangles relational and objective features through relative cross-attention.

3. The authors have proven, through comprehensive experiments, that the Abstractor is more sample-efficient than the vanilla Transformer and previous architectures used for pure or partial relational tasks.

**Weaknesses:**

1. The overly constrained setting limits the significance of the work. The authors conduct experiments under two settings: "purely relational" and "partially relational". As per the authors' definition, "purely relational" implies that object-level features are extraneous and that the statistics of relation/order are already sufficient for solving the task. This is an extremely restricted setting and may not fully represent the complexity of real-world tasks where both relational and object-level information are often important. The authors use math problem-solving to represent the "partially relational" setting, but the math problem here is arguably more relational/symbolic than object-level. It seems like the math problems here can be solved by symbolic rules.

2. Scalability is one of the most significant advantages of Transformer architectures. The performance can increase with the model size and data size. Given that the Abstractor is a variant of the Transformer, it's essential to determine whether the scaling law still applies to the Abstractor. From the results provided, the performance can outperform the vanilla Transformer when the data size increases from the 1000 - 5000 range. But what about using more data and a larger model size? Will the Abstractor consistently improve and outperform the Transformer?
3. Why did the authors choose to replace value vectors with input-independent vectors, while keeping the query and key vectors the same (Q -- X, K -- X, V -- S)? Would not the configuration (Q -- S, K -- S, V -- X) also disentangle object-level features (x) and symbolic vectors (s)? To me, the latter one is more intuitive: the relation weight R_{ij} between i, j is represented by the inner product of symbolic vectors, and then object-level features are weighted by R_{ij}.

**Questions:**

see weakness.

---

> ### Author Response · Authors · 2023-11-22
>
> > The overly constrained setting limits the significance of the work. The authors conduct experiments under two settings: "purely relational" and "partially relational". As per the authors' definition, "purely relational" implies that object-level features are extraneous and that the statistics of relation/order are already sufficient for solving the task. This is an extremely restricted setting and may not fully represent the complexity of real-world tasks where both relational and object-level information are often important.
>
> The importance of purely relational reasoning is suggested by the use of Raven's progressive matrices (for e.g.) in standard tests of human intelligence. Yet, we agree that "purely relational" tasks are not representative of the complexity of real-world tasks, for which both relational and object-level information are often important. Our experiments aim to evaluate the proposed architecture on more controlled purely relational tasks (section 4.1 & 4.2) as well "partially-relational" tasks which are more representative of the complexity of real-world tasks (section 4.3).
>
> The motivation of the experiments on purely relational tasks is two-fold: 1) validate that the Abstractor does indeed achieve improved relational representation in a controlled setting with fewer confounding factors (e.g., relevance of object-level features), and 2) compare the Abstractor to existing relational architectures, which have so far focused on discriminative tasks in which there exists a latent set of relations that form a sufficient statistic (i.e., purely relational). Our main claim about the Abstractor is that it is able to learn good representations of relations. To test this claim, and isolate for the effect of the quality of relational representations, we believe more controlled purely relational tasks are appropriate. The experiments of section 4.1 and 4.2 attempt to do this for discriminative tasks and sequence-to-sequence tasks, respectively.
>
> > The authors use math problem-solving to represent the "partially relational" setting, but the math problem here is arguably more relational/symbolic than object-level. It seems like the math problems here can be solved by symbolic rules.
>
> The math problems can of course be solved by symbolic rules, as you mention. But the Transformer and Abstractor are not symbolic programs. Their performance on such a task is a measure of their representational capacity and sample efficiency for modeling this kind of reasoning. Moreover, it is a longstanding problem to understand how neural computation (e.g., the human brain) carries out apparently symbolic computation such as the mathematical processing tasks in these experiments.
>
> The math problems are character-level sequence-to-sequence tasks, which require modeling both object-level features and relational features. The Transformer and Abstractor-based models are both neural architectures, which must learn to process the input as vector representations, integrating object-level and relational features. In our experiments, we observe that the incorporation of an Abstractor module into a neural architecture for this task yields meaningful performance benefits.

---

> ### Author Response · Authors · 2023-11-22
>
> > Scalability is one of the most significant advantages of Transformer architectures. The performance can increase with the model size and data size. Given that the Abstractor is a variant of the Transformer, it's essential to determine whether the scaling law still applies to the Abstractor. From the results provided, the performance can outperform the vanilla Transformer when the data size increases from the 1000 - 5000 range. But what about using more data and a larger model size? Will the Abstractor consistently improve and outperform the Transformer?
>
> In terms of computational scalability, it is relevant to point out that the Abstractor has the same advantages and disadvantages as a Transformer, due to the similarity in implementation. In particular, future advances in parallel computation and optimization of attention in Transformers can be directly applied to the Abstractor. We agree that testing scaling behavior on Abstractor-based models would be an interesting future direction of research.
>
> In the object-sorting experiments of section 4.2, the task is saturated by the Abstractor at about 500 samples, and eventually saturated by the Transformer at about 3000 samples. Testing scaling of the Abstractor will require more complex tasks (e.g., language modeling) and considerable compute resources, which are out of scope for this project.
>
> > Why did the authors choose to replace value vectors with input-independent vectors, while keeping the query and key vectors the same (Q -- X, K -- X, V -- S)? Would not the configuration (Q -- S, K -- S, V -- X) also disentangle object-level features (x) and symbolic vectors (s)? To me, the latter one is more intuitive: the relation weight R_{ij} between i, j is represented by the inner product of symbolic vectors, and then object-level features are weighted by R_{ij}.
>
> Thanks for the question. We hope the following will help to clarify.
>
> Consider first the case of symmetric relations. Given a pair of objects $x, y$, we think of $\langle \phi(x), \phi(y) \rangle$ as a relation between $x$ and $y$ in the following sense: 1) $\phi$ extracts features of the objects, and 2) the inner product compares the two features. The inner product will be large when the two feature vectors are similar, and small otherwise. In the case of asymmetric features, the feature extractor for each object can be different. With this in mind, relational cross-attention can be thought of as a form of message-passing where the message from object $j$ to object $i$ is $m_{j \to i} = (s_j, r(x_i, x_j))$, encoding the relation between the two objects and a symbol identifying the sender. In relational cross-attention, the symbols act as identifiers of the objects, but do not encode the objects' features.
>
> In the configuration $(Q \gets S, K \gets S, V \gets X)$, the inner products would be $\langle s_i, s_j \rangle$. Since the set of symbols $S$ only represent the identity of the objects and not their features, the inner products between them would not represent relations between the input objects (in fact, the relations would be constant in the case of positional symbols). Moreover, since the values are the input objects ($V \gets X$), the object-level features are propagated through, and the relational bottleneck is not obeyed.
>
> ---------------------
>
> Many thanks for your thoughtful engagement with our work! We hope this has addressed some of your questions and concerns. Please let us know if you have any other questions.

---

> > ### Comment · Reviewer_4fms · 2023-11-23
> > **Response to authors**
> >
> > Thank you for the clarification. I will keep the positive score.

---

### Official Review · Reviewer_nkpJ · 2023-10-31

**Soundness:** 3 good
**Presentation:** 3 good
**Contribution:** 3 good
**Rating:** 6
**Confidence:** 3

**Summary:**

This paper proposes a novel Abstractor module that uses relational cross-attention to enable explicit relational reasoning. such relational cross-attention disentangles relational information from object-level features. The experimental results show that such explicit relational reasoning greatly improves the sample efficiency of Transformers on relational tasks.

**Strengths:**

1. The paper is well-written and easy to follow.
2. The Abstractor module and relational cross-attention are novel and interesting, simple but effective.
3. The method is well motivated and the proposed method is well justified on a variety of tasks.

**Weaknesses:**

1. When comparing sample efficiency on pure relational tasks, the Abstractor is only compared to the Transformer baseline. It would be better to show the comparison with other relational structures like PrediNet.
2. There is no explicit section for related work and limitations are not discussed.

**Questions:**

1. Could you compare both the performance and sample efficiency of Abstractor with other relational structures like PrediNet?
2. STSN (Mondal et al., 2023) has used Transformer for RAVEN and PGM problems, which involve relational reasoning, how much would the Abstractor improve over Transformer in those tasks?

---

> ### Author Response · Authors · 2023-11-22
>
> Thank you for taking the time to read our work and provide this feedback.
>
> > Could you compare both the performance and sample efficiency of Abstractor with other relational structures like PrediNet?
>
> Thank you for the suggestion! We agree that comparing to PrediNet is an important baseline for the discriminative relational tasks in addition to CoRelNet. We have added this baseline to the paper. On the pairwise order relation task, we observe that while PrediNet performs better than an MLP and the standard symmetric variant of CoRelNet, it struggles compared to the Abstractor and even the asymmetric variant of CoRelNet. On the SET task, however, PrediNet performs better than CoRelNet, although it still lags behind the Abstractor in terms of sample efficiency. This might be explained by the fact that, unlike CoRelNet, PrediNet is able to represent multiple relations simultaneously---hence it performs better on SET which requires modeling multi-dimensional relations. The superiority of the Abstractor may be due to inner products being a better inductive bias for modeling relations than the difference comparators used in PrediNet (e.g., inner products obey a stricter relational bottleneck).
>
> > STSN (Mondal et al., 2023) has used Transformer for RAVEN and PGM problems, which involve relational reasoning, how much would the Abstractor improve over Transformer in those tasks?
>
> This is an interesting empirical question. We leave this to future work.

---

> > ### Comment · Reviewer_nkpJ · 2023-11-23
> >
> > Thanks for answering questions and providing new experimental results about PrediNet.

---

### Official Review · Reviewer_eiad · 2023-10-31

**Soundness:** 4 excellent
**Presentation:** 4 excellent
**Contribution:** 3 good
**Rating:** 8
**Confidence:** 4

**Summary:**

The authors introduce a modification of self-attention, referred to as "Relational Cross-Attention" (RCA), in which value projections are replaced with input-independent vectors, referred to as "symbols".  These symbols are either learned or implemented as (relative) positional embeddings. Additionally, the softmax function used in the attention operation can be replaced with element-wise activation functions for improved performance on some tasks. The authors replace the attention-layer in a transformer encoder layer with RCA, and refer to the resulting layer as an _Abstractor Module_.

Various experiments on toy-problems requiring relational reasoning are carried out. The authors show that Abstractors out-perform transformers and CoRelNets (another explicitly relational baseline) on these tasks, and that Abstractors are able to learn generalizable relations (demonstrated through pre-training experiments) and can easily be constrained to learn only symmetric relations (through symmetric inner products $\phi_{\theta}(x)^\top \phi_{\theta}(x)$, rather than the default $\phi_{\theta}(x)^\top \psi_\omega(x)$ in RCA). Additionally, they experiment with various forms of overall architecture, varying whether the Abstractor layer is used alongside Encoders (in parallel or in sequence) prior to decoding.

**Strengths:**

Overall presentation is very clear and the method is well-explained, though the analogy to self-attention may be better mentioned earlier in the paper, rather than first "deriving" the relationship between the calculation of a project relation matrix and $QK^\top$ Self-Attention.

The experiments serve to demonstrate the efficacy of Abstractors in toy settings, and some effort is made to demonstrate that they do indeed learn re-usable relations which are robust to noisy data. Alongside the ablations, these provide sufficient evidence that abstractors provide a strong relational inductive bias which may make them useful in more applied settings - though this remains to be demonstrated.

The experiments comparing Abstractors with pre-learned relations (per-head) against a symbolic input MLP are particularly well-thought-out and interesting, in that they go some way toward demonstrating the generality of the learned relations.

**Weaknesses:**

Methodologically there are no substantial weaknesses, as comparisons against baselines are made as fair as possible. One minor exception is the omission of a baseline transformer for the Order Relation and SET tasks, though this was presumably an intentional omission based on the already superior performance of CoRelNet compared to Transformers?

There is one consistently made claim which may be slightly overstated (this is a question to the authors) - namely that the output of Abstractors represents "purely relational information"; I believe this only holds if there are no residual connections in the abstractor module implementation being used (which is offered as an option); if there is a residual connection, then it seems the abstractor MLP could still learn to operate on information present in object-level representations.

Related to the above point, it would be very interesting to see an attempt to "lift" the learned relational information into a symbolic form, as was done e.g. in the cited PrediNet work (Shanahan)

**Questions:**

Some minor questions:
1. Multi-attention decoder: Perhaps this form of decoder is standard, but why is CausalSelfAttention used before Cross-Attention (which is not causally masked?)
2. On the Math Problem experiments: It seems the dataset consists of 8 tasks, but only 5 of these are investigated in the experiments. Is there a particular reason for this omission?
3. On the SET Comparison against a symbolic-input model: When pre-training the abstractor relations, is the input from the same pre-trained CNN as when training the multi-head abstractor, or also from the symbolic inputs?
4. The authors argue that relations are well-modelled as inner-products. I am curious as to which differences this might impose on the learned relations when compared to the relational-form used in the PrediNet, in which a difference of projections ("differential comparator") is used?

---

> ### Author Response · Authors · 2023-11-22
>
> Thank you for your thoughts and feedback! We are glad you found our work interesting.
>
> > One minor exception is the omission of a baseline transformer for the Order Relation and SET tasks, though this was presumably an intentional omission based on the already superior performance of CoRelNet compared to Transformers?
>
> Indeed, relational architectures (like CoRelNet) tend to perform better than Transformers on such discriminative relational tasks. Though, so far, relational architectures have not explicitly considered the generative or sequence-to-sequence setting. For this reason, for the discriminative tasks, we decided to focus our comparison to existing relational architectures, whereas for sequence-to-sequence tasks we focus our comparison on Transformers.
>
> Of relevance here, one relational architecture we did not include initially as a baseline is the PrediNet architecture (Shanahan et al.). We have now added it to the list of baselines for the discriminative experiments. For completeness, we also added an MLP baseline (as expected, this performs worse than the relational architectures). You can find these results in the updated pdf.
>
> > There is one consistently made claim which may be slightly overstated (this is a question to the authors) - namely that the output of Abstractors represents "purely relational information"; I believe this only holds if there are no residual connections in the abstractor module implementation being used (which is offered as an option); if there is a residual connection, then it seems the abstractor MLP could still learn to operate on information present in object-level representations.
>
> The residual connection adds the *abstract states* from the previous layer $A^{(l-1)}$, not the object representations $X$. Hence, even with a residual connection, the information represented in $A^{(l)}$ is still purely relational---albeit now incorporating relational information at multiple layers of hierarchy.
>
> > Multi-attention decoder: Perhaps this form of decoder is standard, but why is CausalSelfAttention used before Cross-Attention (which is not causally masked?)
>
> The paper "Exploring the Limits of Transfer Learning with a Unified Text-to-Text Transformer" by Raffel et al. has some discussion on causal masking which might be useful (e.g., see section 3.2.1, Fig 3, and Fig 4. Autoregressive encoder-decoder models are depicted in the left-hand side of Fig 4). In an autoregressive Encoder-Decoder architecture, the input to the decoder is the right-shifted target sequence, and causal masking prevents attending to object $j > i$ when predicting the target at $i$. This allows for an encoder-decoder architecture to autoregressively produce an output sequence. But, the entire "context sequence" in the encoder can be attended to by the decoder, so there is no need for masking cross-attention. For example, in the case of the math seq2seq task, when predicting the $i$-th token in the solution (output sequence) the model can attend to all information in the question (input sequence), as well as to the portion of the solution (output sequence) generated so far (i.e., up until token $i$).
>
> Please let us know whether this answers your question.

---

> > ### Comment · Reviewer_eiad · 2023-11-22
> >
> > Thank you for your detailed response and the changes. The inclusion of PrediNet as an additional baseline on the discriminative experiments is particularly nice to see.
> >
> > The clarification regarding the way in which residual connections are used is also valuable - as such, this minor complaint has been addressed as a misunderstanding on my part.

---

> ### Author Response · Authors · 2023-11-22
>
> > On the Math Problem experiments: It seems the dataset consists of 8 tasks, but only 5 of these are investigated in the experiments. Is there a particular reason for this omission?
>
> The math dataset contains 8 "modules", but each module contains several tasks. The total number of tasks is quite large. We did not have the computational resources to run an evaluation on all tasks. The tasks were chosen as a representative set of the full set of tasks.
>
> > On the SET Comparison against a symbolic-input model: When pre-training the abstractor relations, is the input from the same pre-trained CNN as when training the multi-head abstractor, or also from the symbolic inputs?
>
> When pre-pretraining the Abstractor relations for this experiment, the inputs are the embeddings generated by the pre-trained CNN. In contrast, the MLP receives as input a hard-coded binary representation of the four latent relations. The idea behind this comparison is to evaluate the quality of the geometry of the representations produced by an Abstractor. The hard-coded binary representation of relations is an "ideal" representation, in the sense that it is completely disentangled and noise-free. The very small difference in learning curves between the MLP with hand-coded symbolic representations and the Abstractor is an indication of the quality of representational geometry produced by the Abstractor.
>
> > The authors argue that relations are well-modelled as inner-products. I am curious as to which differences this might impose on the learned relations when compared to the relational-form used in the PrediNet, in which a difference of projections ("differential comparator") is used?
>
> A very interesting question. Here are some thoughts on this based on our understanding of the PrediNet paper.
>
> The PrediNet paper argues for a particular philosophy based on propositional logic about how knowledge and relations ought to be represented. But in terms of representations, the architectures consists of the following steps, 1) retrieve a pair of objects (one pair for each 'head'), 2) project the object representations into a one dimensional space (one for each relation), and 3) take the difference between those scalars. This produces a vector of `n_pairs x n_relations` entries (modulo some additional entries encoding position, since in their paper they consider the input to be an image). At its core, as you said, PrediNet represents a relation as a difference between one-dimensional projections whereas we represent relation as an inner product between two projections onto $d$-dimensional space.
>
> In some sense, differences and inner products are similar in terms of information content. For example, if $x, y$ are unit-length, $\langle x, y \rangle$ is proportional to $\lVert x - y \rVert^2$. A couple of simple differences to begin with. Also, the PrediNet architecture computes relations by first retrieving a pair of objects via an attention mechanism, whereas we consider all possible pairs.
>
> One advantage of inner products over differences as representations of relations is greater "invariance to representation". For example, applying an orthogonal transformation to a pair of vectors does not change the inner product, but would change the difference (though not the norm of the difference). Moreover, inner products have certain robustness properties that may be useful for learning representations (e.g, for a random matrix $\Phi$ with iid Gaussian entries, $\langle \Phi x, \Phi y\rangle \approx \langle x, y \rangle$; Zhou et al. (2009)). We explore some of this in section C.1 of the appendix, where we present some experiments on robustness to corruptive noise of different forms.
>
> These are some of our initial thoughts on the topic. Indeed, with the addition of PrediNet as a baseline for the experiments in section 4.1, we observe that the Abstractor tends to perform better. But the PrediNet architecture includes some added confounders. It may be interesting to explore the differences between inner product relations and difference relations in a more controlled setting (e.g., by creating a simplest possible architecture with this inductive bias, similar to what CoRelNet aimed to do).

---

> ### Comment · Reviewer_eiad · 2023-11-22
>
> I agree with your summary of the PrediNet approach, and your discussion of the potential reasons why (as it turns out) inner products may be superior to differential comparators as used in the PrediNet. Indeed, these invariance is particularly interesting as the PrediNet would need to dedicate different rows of $W_S$ to the computation of the same relation under some orthonormal transformation of its inputs. Thank you for your well-thought-out response to this!
>
> I will maintain my high-confidence score from before as there were no substantial weaknesses, but minor weaknesses or misunderstandings have now been clarified too.

---

### Author Response · Authors · 2023-11-22

We would like to extend our thanks to all reviewers for taking the time to engage with our work and provide feedback.

We made a few additions to the paper since the initial submission that we'd like to share with you. We've uploaded the updated pdf. In particular, we would like to draw your attention to the following:
- We've included additional baselines to the discriminative relational tasks experiments (section 4.1), including PrediNet and MLP baselines.
- We've extended the Abstractor framework by proposing several "symbol assignment" mechanisms, with different properties. This is described in section 2.3 "Symbol Assignment Mechanisms." As you know, the first step in the Abstractor model is to "assign a symbol to each object." In the previous version of the paper, we considered (what we are now calling) "positional symbols," where symbols are assigned to objects sequentially based on the order they appear. While simple, this method has the advantage of strictly obeying the relational bottleneck and it works well in our experiments. In section 2.3, we propose two other symbol assignment mechanisms: position relative symbols (which we mentioned in the previous version of the paper as well), and "symbolic attention". Symbolic attention enables symbols to come to represent information about the object's 'syntactic role', which can be shared across problem instances. That is, the symbol identifies an object in relational cross-attention based on its role, rather than only its position.
- In the math experiments of section 4.3, we've added an evaluation of an Abstractor model with symbolic attention. In the previous version, we compared an Abstractor with positional symbols to a standard Transformer, and observed consistent improvements. The Abstractor with symbolic attention yields a further improvement.

We hope you will look at these additions and provide any comments or questions you have. In addition, we have responded to each reviewer's individual comments.  We look forward to hearing your thoughts.

---

### Meta-Review · Area_Chair_zvqZ · 2023-12-08

**Metareview:**

This submission contributes an attentional mechanism suited to capture relation information across elements as opposed to specific values. The submission generated many solid discussions and was seen as an interesting addition to the literature. The reviewers appreciated the clear presentation and the fair and conclusive comparison with baselines, though more baselines would have been welcomed.

**Justification For Why Not Higher Score:**

Only one reviewer gives a really strong rating

**Justification For Why Not Lower Score:**

All reviewers are positive. In addition, the manuscript is clear with fair comparisons. Such manuscripts contribute to a solid scientific literature.

---

### Decision · Program_Chairs · 2024-01-16

Accept (poster)